# MultiON: Benchmarking Semantic Map Memory using Multi-Object Navigation

**Saim Wani**[1]* **Shivansh Patel**[1,3]* **Unnat Jain**[2]* **Angel X. Chang**[3] **Manolis Savva**[3]

[1]IIT Kanpur     [2]UIUC     [3]Simon Fraser University

https://shivanshpatel35.github.io/multi-ON/

## Abstract

Navigation tasks in photorealistic 3D environments are challenging because they require perception and effective planning under partial observability. Recent work shows that map-like memory is useful for long-horizon navigation tasks. However, a focused investigation of the impact of maps on navigation tasks of varying complexity has not yet been performed. We propose the *multiON* task, which requires *navigation to an episode-specific sequence of objects in a realistic environment*. MultiON generalizes the ObjectGoal navigation task [2, 52] and explicitly tests the ability of navigation agents to locate previously observed goal objects. We perform a set of multiON experiments to examine how a variety of agent models perform across a spectrum of navigation task complexities. Our experiments show that: i) navigation performance degrades dramatically with escalating task complexity; ii) a simple semantic map agent performs surprisingly well relative to more complex neural image feature map agents; and iii) even oracle map agents achieve relatively low performance, indicating the potential for future work in training embodied navigation agents using maps.

## 1 Introduction

Recent work on embodied AI agents has made tremendous progress on tasks such as visual navigation [25, 39, 52], embodied question answering [20, 44], and natural language instruction following [3]. This progress has been enabled by the availability of realistic 3D environments and software platforms that simulate navigation tasks within such data [32, 38, 39, 47]. Large-scale training has led to near-perfect agent performance for basic visual navigation tasks under certain assumptions [45].

At the same time, memory has emerged as a key bottleneck for further progress in longer-horizon tasks. Embodied navigation agents are surprisingly brittle, with complete failure being common when the path to be navigated and number of locations to be visited is modestly increased [3, 39, 45]. In response, much prior work has incorporated some form of map-like memory to aid performance in longer-horizon tasks [8, 15, 22, 25–27, 35, 37, 51]. This prior work has mostly focused on proposing approaches for constructing the maps from partial observations of the environment, and architectures for leveraging the map representation. Though map-like memory structures need not be optimal for learning-based agents, they bring the advantage of a widely-used spatial abstraction and human interpretability. They also impose inductive bias tied to the structure of interiors that have been shown to outperform implicit memory architectures in a variety of navigation tasks [7, 13–15].

We focus on studying *what* information is useful in maps, under perfect agent localization and map aggregation. This is in contrast with prior work which focuses on *how* to aggregate information under

imperfect localization [15, 27]. Perfect localization is a simplifying assumption that decouples the question of what is useful in a map representation from dealing with noisy sensing and actuation.

We propose *multiON*, a task framework that involves *navigation to an ordered sequence of objects* placed within realistic 3D interiors. This task framework generalizes earlier visual navigation tasks such as the PointGoal and ObjectGoal tasks [2], by defining an arbitrary sequence of semantically distinct objects as navigation goals. Using a set of multiON experiments, we benchmark agents with egocentric mapping, agents using oracle maps, and agents possessing no map memory.

The multiON framework allows for direct control of task difficulty by generating navigation episodes with multiple goal objects. At the same time, the base navigation task is simple, allowing us to focus our investigation on map utilization in visual navigation. Though there is much prior work involving higher-level understanding, planning and reasoning (e.g., natural language understanding [3, 12, 28], embodied question answering [20, 50], interaction with the environment [41], leveraging world priors [49]) we avoid introducing these elements to limit confounding factors that influence the impact of map memory on visual navigation. In summary, we make the following contributions:

- Propose the multiON task framework to allow for controlled, systematic investigation of visual navigation in 3D environments.
- Benchmark the impact of egocentrically-constructed map representations and oracle maps on navigation performance across a breadth of navigation task complexities.
- Show that agent navigation performance drops dramatically with task complexity, that a simple semantic map construction approach outperforms a more complex neural image feature map agent, and that even oracle map agents achieve relatively low performance, indicating the potential for future work in this direction.

## 2    Related Work

**Embodied AI agents.**  There has been much interest in studying AI agents in simulated 3D environments [1, 9, 11, 32, 47, 4, 39, 48, 43]. Learning how to tackle a variety of tasks from egocentric perception is a common theme in this area. Embodied navigation is a family of closely related tasks where the goal is to navigate to specific points, objects or areas, respectively PointGoal, ObjectGoal, and AreaGoal [2]. In PointGoal navigation [39, 14, 23], the agent has access to a displacement vector to the goal at each time step, largely obviating the need for long-horizon planning and map-like memory. PointGoal has been extensively studied and recently 'solved' [45]. In contrast, ObjectGoal has not been well-studied despite being introduced in early work by Zhu et al. [53] and explored in natural language grounding settings [12, 28].

We generalize the ObjectGoal navigation task to allow systematic investigation of agents using spatial and semantic map representations for grounding egocentrically-acquired visual information.

**Long-horizon embodied agent tasks.**  There has been relatively little work on training embodied agents for long-horizon tasks. Mirowski et al. [33] propose an LSTM-based architecture for a *courier task*, where an agent navigates to a series of random locations in a city. Multi-target Embodied QA [50] extends single object EQA [19] to answer questions such as "Does the table in the bedroom have same color as the sink in the bathroom?", which require visually navigating to exactly two objects. Fang et al. [21] propose a transformer-based architecture for a variety of tasks (collision avoidance, scene exploration, and object search), of which object search is most related to our work. In object search the agent navigates to televisions, refrigerators, bookshelves, tables, sofas, and beds in the scene with no specific order. Note that the position and number of the objects within the scene is not controlled, and thus the complexity of the task is fixed and a property of the dataset. Beeching et al. [7] use spatial memory for several tasks including an 'ordered k-item' task [6] within a gridworld-like game environment [31]. The item list is kept fixed across all experiments.

Our multiON task requires navigation to an episode-specific ordered list of objects within a challenging photorealistic 3D environment. The episode-specificity requires grounding object class labels to their visual appearance. The ordered aspect requires storing information on previously encountered objects that need to be retrieved later (e.g., seeing the third object on the way to the first one). In contrast, 'ordered k-item' [6] is not episode-specific and 'object search' [21] is neither episode-specific nor ordered (see Table 1). To the best of our knowledge, multiON is the first task designed to benchmark long-horizon embodied navigation in photo-realistic environments.

Table 1: Comparison of multiON to navigation tasks with multiple object goals from prior work. Note that prior work does not adopt a FOUND action, so incidental navigation to a goal is treated as success. Moreover, the set of objects is held fixed, with no episode-specifity for the goal objects.

| Task | Environment | Position control | # Objects | Goal | Evaluation |
|------|-------------|:----------------:|-----------|------|------------|
| Search [21] | Synthetic | ✗ | 6 | all in any order | reward, classes found |
| Ordered $k$-item [6] | VizDoom [31] | ✓ | 4 or 6 | fixed set, in order | reward |
| MultiON (ours) | Habitat [39] | ✓ | arbitrary | episode-specific ordered set | success & efficiency metrics |

**Maps.** Maps which encode semantic information about the environment broadly fall into two categories: spatial [25, 35] and topological [15, 37]. *Spatial maps* are tensors where two dimensions align with an environment's top-down layout. Other dimensions contain features encoding semantic information for a particular location of the environment. Spatial maps built with Simultaneous Localization and Mapping (SLAM) have been used for tasks such as exploration [51], and playing FPS games [8]. Work on deep RL for embodied navigation has leveraged egocentric maps [25], and a combination of egocentric and allocentric maps [26]. Gordon et al. [22] store object detection probabilities in spatial maps to aid recognition for question answering. More recent work investigates learning and use of semantic grid-based maps for navigating to a single ObjectGoal [13, 10]. Chaplot et al. [13] perform first person view semantic segmentation and project the segmentation onto the top-down map using a differentiable mapping module. Cartillier et al. [10] study whether it is better to segment and then project, or to project and then segment directly on the top-down map. These works assume perfect knowledge of agent egomotion (i.e. perfect localization). MapNet [27] does not make this assumption and integrates information over time for use in an end-to-end differentiable architecture. In contrast, *topological maps* such as those proposed by Chaplot et al. [15], Savinov et al. [37] do not align visual information to the environment's top-down layout. Instead, they store landmarks (e.g., specific input frame) as nodes and their connectivity as edges. Additionally, prior knowledge of scene layout is used as a knowledge graph in various works [49, 46].

Despite this rich body of work using maps for embodied AI, there has been no systematic investigation of what map information benefits the core task of navigation, and how much value egocentrically acquired maps add relative to oracle maps or no map representations. We focus on these questions in this paper.

## 3 Task

Here we define the multiON task in detail. In an episode of multiON, the agent must navigate to an ordered sequence of objects $G$ placed within the environment, where $G_n$ is the $n$-th object in the sequence. The number of objects $m$ allows us to vary the complexity of the navigation episode and create more 'long-term' navigation scenarios. We use $m$-ON to refer to an episode with $m$ ordered goal objects. The $m$ objects are selected from a set of $k$ available objects where $k \geq m$.

During the episode, the agent receives at each step $t$: i) egocentric RGB and depth sensor images $o_t$; and ii) current goal object specified as a $k$-dimensional one-hot vector $g_t$. The agent must navigate to the vicinity of the current goal object $G_n$ and declare the object has been found using a FOUND action. The goal object specification is then updated, and the agent must continue to locate the next object $G_{n+1}$ in the sequence $G$ until all objects have been located. If the FOUND action is called while not in the vicinity of the current goal object, or if the agent has taken a threshold maximum number of actions the episode is declared a failure. Otherwise, the episode is successful.

The multiON task is a generalization of the ObjectGoal navigation task proposed by Zhu et al. [53] and Anderson et al. [2] allowing more controlled complexity through the selection of multiple object goals. The flexibility of multiON allows us to systematically study the benefit of map memory. To summarize, some of the key differentiating characteristics of our task are:

- A different goal sequence is specified to the agent per episode necessitating grounding of object identity to visual inputs, in contrast to Fang et al. [21], Beeching et al. [6] where the goals are fixed and predetermined.
- The goal objects are inserted into realistic 3D environments at controlled locations. Programmatic placement and category selection of the objects allows us to precisely control the

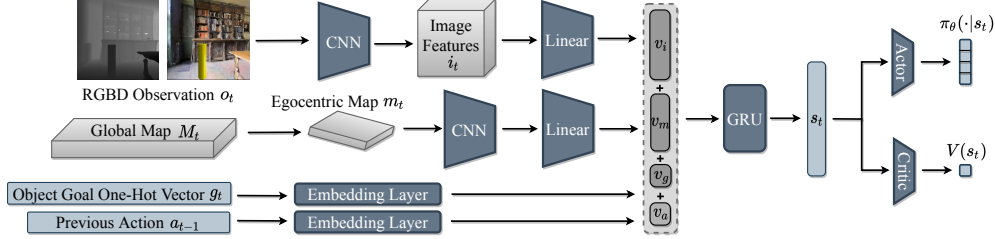

Figure 1: **Base agent architecture.** At each time step $t$, RGBD frames $o_t$, an egocentric map $m_t$, the object goal $g_t$, and the previous action $a_{t-1}$, are encoded into a concatenated embedding $[v_t, v_m, v_g, v_a]$ passed to a GRU to construct state $s_t$ for an actor-critic pair used by the agent.

geodesic distances between target objects and the resulting episode complexity (in contrast to relying on pre-existing objects [21, 13]).
- The agent needs to explicitly declare when the goal object has been FOUND, in line with the recommendation of Anderson et al. [2].
- Task evaluation metrics are defined independent of reward formulations, allowing future work to explore different reward structures and other methods such as imitation learning.

In this paper, we adopt the following specific instantiation of multiON. The agent is embodied with a cylindrical body of radius 0.1m and height 1.5m. The agent action space has four actions: {FORWARD, TURN-LEFT, TURN-RIGHT, FOUND} where turns are by $30°$ and FORWARD moves forward by 0.25m. An $m$-ON episode specifies the agent start position and orientation, and the positions of $m$ goal objects. The objects are randomly sampled without replacement from a set of 8 cylinder objects with height 0.75m and radius 0.05m and different colors: {red, green, blue, cyan, magenta, yellow, black, white}. As such, there are $\binom{8}{m} \cdot m!$ possible goal sequences $G$.

## 4 Agent Models

We focus on examining the benefit of incorporating spatial maps for embodied navigation agents. We first introduce a general network architecture that is shared by all the agent models we study. Then, we specify variations on this base architecture for each model, comparing agent models with no map representation to ones with access to a ground truth map, as well as agents with learned maps.

### 4.1 Base Agent Architecture

**Inputs.** At each time step the agent receives a number of sensory inputs. These include a visual observation $c_t$ (RGB image of $256 \times 256 \times 3$), a corresponding $256 \times 256 \times 1$ depth observation $d_t$, and the object goal specified as a one-hot vector $g_t$. The agent also has access to the action at the previous step $a_{t-1}$, a common strategy for augmenting purely reactive agents [38, 20, 42]. We denote the agent's sensory observations as $o_t = (c_t, d_t)$. All agents use either an explicit spatial grid map representation, or have implicit memory: the NoMap baselines (RNN, SMT, FRMQN). We refer to the map of the entire environment as the 'global map' and denote it with $M$. The agent models operate with an egocentric map $m_t$, representing a rotated and cropped view of the global map centered and oriented at the agent's location and orientation at step $t$. Figure 1 illustrates this agent architecture.

**Network components.** The RGBD sensory input $o_t$ is transformed into image features $i_t$ and linearly embedded into $v_i$ using a CNN block and a linear layer. Similarly, the egocentric map $m_t$ is embedded into $v_m$. These linear embeddings are then concatenated with 16-dimensional embeddings $v_g$ and $v_a$ for the current goal definition $g_t$ and previous action $a_{t-1}$. The concatenated representation $v_t = \text{concat}(v_i, v_m, v_g, v_a)$ is transformed using a Gated Recurrent Unit (GRU) [18] to give the final state representation $s_t$ at step $t$. The GRU provides an implicit memory over the embedded inputs. Let $\theta$ represent the parameters of this end-to-end trainable neural architecture. The output of the GRU is used to predict the action policy $\pi_\theta(\cdot|s_t)$ and the approximate value function $V(s_t)$. We next describe the critical difference between agent models which lies in how the global map $M$ is constructed and utilized.

## 4.2 Agents

We investigate the performance of different agents on the multiON task. We select representative agents with implicit memory (`NoMap(RNN)`, `FRMQN` [34], `SMT` [21]) and compare them against agents that use explicit grid map representations, with both oracle maps and learned maps. In addition, we consider two random action baselines, with one variant selecting the FOUND action randomly, and the other using oracle FOUND.

**Implicit memory baselines**

`NoMap(RNN)`: An agent that does not use explicit map information. This agent establishes a comparison point for the performance of our base architecture without maps as trained through PPO [40].

`FRMQN`: This agent is based on Oh et al. [34]'s architecture which stores observation embeddings of the previous episode steps in an LSTM-like memory as key-value pairs, and reads from this memory using soft attention.

`SMT`: This agent is based on Fang et al. [21]. Here, an embedding of each observation is again stored in memory, and an encoder-decoder network is used to retrieve relevant information from the memory. We use the same CNN as in our other agent models to produce the observation embeddings.

**Oracle baselines**

`OracleMap`: This agent gets the entire global ground truth (i.e. oracle) map $M_{\text{GT}}$ for the current environment. The global map is a $300 \times 300$ cell grid with $0.8\text{m} \times 0.8\text{m}$ cells. Each cell contains two channels: i) occupancy information; and ii) goal object category. Concretely, the occupancy channel stores a 16-dimensional learnable neural embedding of whether a cell is navigable, non-navigable or outside the scene. The goal object category channel also stores a 16-dimensional embedding corresponding to the category of the object occupying that cell [36, 24]. There are $k + 1$ categories: $k$ goal categories, and a 'no goal' category. This oracle map agent provides an upper bound for the benefit of the spatial map to agent navigation performance in multiON.

`OracleEgoMap`: Same as `OracleMap`, except that the global map is progressively 'revealed' from an egocentric perspective as the agent explores the environment. Thus, at step $t$ the agent has access to a partially revealed global map $M_t$. Note that visibility is accounted for by ray tracing and checking against the agent's current depth frame $d_t$ so that the agent cannot see behind obstacles or outside its field of view. Please refer to Appendix A.2 for details. This agent represents an idealized egocentric mapping agent (i.e. perfect sensing of the environment within a limited field of view)

**Learned map agents**

`ObjRecogMap`: This agent does not use any oracle map information. It progressively constructs a global map storing predicted categories of goal objects visible in the sensory inputs. An auxiliary $(k + 1)-$way classification network is supervised via the ground truth object category at train time to predict which goal object is in view. If no goal is within the field-of-view ($79°$) and $2.5$m of the agent, a 'no object' label is assigned. The embedding of this predicted category is used to populate the global map $M_t$ at the grid cell corresponding to the agent's position. See Appendix A.2 for details about this auxiliary classification task. This agent is representative of approaches that embed egocentrically observed object information into a spatial map. Surprisingly, to our knowledge there is no prior work employing this straightforward approach. Our scheme is similar to the one employed by Gordon et al. [22], where object detection probabilities are encoded into a map (in addition to occupancy, coverage, and navigation intent). However, we use a simpler object classifier rather than an object detector (i.e. the output is a label rather than a detection localizing the object).

`ProjNeuralMap`: This agent neurally projects image features to the map using the projection module of the MapNet architecture [27]. Figure 2 illustrates the approach: i) depth-conditioned projection of the image features $i_t$ to generate egocentric map $m_t$; and ii) registering $m_t$ to a global map $M_t$ accumulated over time. Concretely, we use the depth buffer $d_t$ to appropriately project each feature $i_t(i, j, \cdot)$ onto the egocentric grid $m_t$. Note the agent is always at the mid-bottom of $m_t$. This egocentric $m_t$ is integrated into $M_t$ via a registration function $\text{R}(m_t, M_t|p_t)$, where $p_t$ is the agent's position and orientation. We integrate neural features into cells that are already filled using

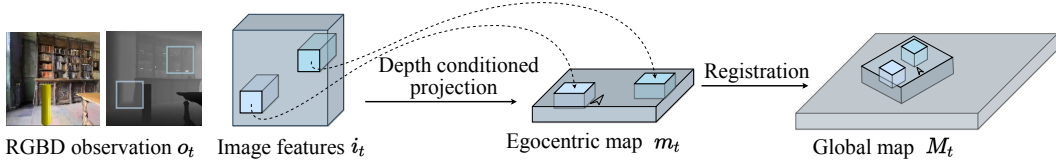

RGBD observation $o_t$    Image features $i_t$    Egocentric map $m_t$    Global map $M_t$

Figure 2: **Projection and map registration module.** Input depth $d_t$ and color $c_t$ frames are encoded into image features $i_t$ which are then projected into an egocentric map $m_t$. This map is then registered onto the global map $M$.

element-wise max-pooling. Refer to Appendix A.2 for more details. This agent is a representative of approaches storing neural image features in the map.

`EgoMap`: This agent is adapted from Beeching et al. [7], who use projection architecture based on MapNet [27] to construct a semantic map. It differs from `ProjNeuralMap` in using an attention-based read mechanism to read from the semantic map.

**Random baseline agents**

`Rand`: This agent chooses an action randomly from the action space: {FORWARD, TURN-LEFT, TURN-RIGHT, FOUND}. It establishes a navigation performance lower bound and reveals the difficulty of variations of the multiON task with time-limited random search.

`Rand+OracleFound`: Chooses an action randomly from: {FORWARD, TURN-LEFT, TURN-RIGHT}. When the agent is within a threshold distance of the current goal, an 'oracle' calls the FOUND action. This implies the episode terminates either on success, or if the agent takes the maximum allowed steps (2,500 steps for all experiments here).

## 4.3   Training

**Reward structure.** The agent receives the following reward $r_t$ at each timestep $t$:

$$r_t = \mathbb{1}_{\text{[reached-goal]}} \cdot r_{\text{goal}} + r_{\text{closer}} + r_{\text{time-penalty}} = \begin{cases} r_{\text{goal}} + r_{\text{closer}} + \alpha & \text{if goal is found} \\ r_{\text{closer}} + \alpha & \text{otherwise} \end{cases},$$

where $\mathbb{1}_{\text{[reached-goal]}}$ is an indicator variable equal to 1 if a FOUND action was called within the threshold distance of the current target goal in the current time step, $r_{\text{closer}} = (d_{t-1} - d_t)$ is the decrease in geodesic distance to the current goal at timestep $t$ (in meters), $r_{\text{goal}}$ is the reward for discovery of a goal, and $\alpha$ is a negative slack reward that encourages the agent to take shorter paths. We use $r_{\text{goal}} = 3.0$ and $\alpha = -0.01$ for all experiments.

**Training setup.** The agent is trained using the reward structure defined above using proximal policy optimization (PPO) [40]. All the agent models are trained for 40 million frames using 16 parallel threads. We use 4 mini-batches and do 2 epochs in each PPO update. We do not tweak any other hyperparameters. The object recognition network in `ObjRecogMap` is supervised through a cross-entropy loss on the ground truth goal object category.

## 5   Experiments

Here we describe a series of experiments with the various agent models on the multiON task. We seek to answer the following questions: (i) how much does an occupancy-based map help over implicit memory? (ii) does incorporating semantics into the map benefit agent navigation performance? (iii) how does our agent performance compare with prior work on similar navigation tasks?

## 5.1   Datasets

We generate datasets for the 1-ON, 2-ON, and 3-ON tasks on the Matterport3D [11] scenes using the standard train/val/test split. As recommended by Anderson et al. [2] there is no overlap between train, val and test scenes. For each $m$-ON dataset the train split consists of 50,000 episodes per scene, and the validation and test splits each contain 12,500 episodes per scene.

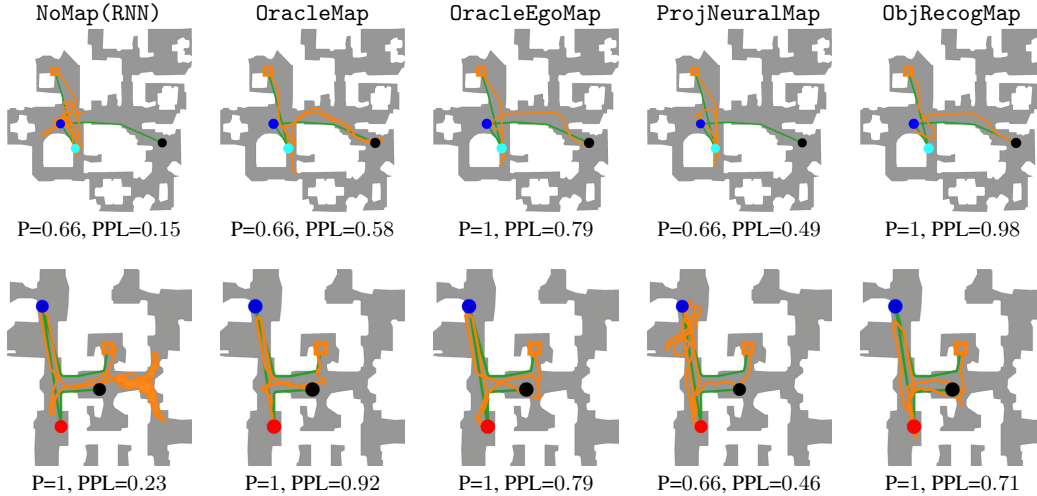

| NoMap(RNN) | OracleMap | OracleEgoMap | ProjNeuralMap | ObjRecogMap |
|---|---|---|---|---|
| P=0.66, PPL=0.15 | P=0.66, PPL=0.58 | P=1, PPL=0.79 | P=0.66, PPL=0.49 | P=1, PPL=0.98 |
| P=1, PPL=0.23 | P=1, PPL=0.92 | P=1, PPL=0.79 | P=0.66, PPL=0.46 | P=1, PPL=0.71 |

Figure 3: Example episodes for different agents. Agent path and shortest path in orange and green, with the start shown by □ (orange square). Goal order for top: 1●, 2●, 3●, and bottom: 1●, 2●, 3●.

The agent starting position and goal locations are randomly sampled from navigable points on the same floor within each environment between which there exists a navigable path. The geodesic distance from the agent starting position to the first goal and between successive goals is constrained to be between 2m and 20m. This ensures that 'trivial' episodes are not generated.

## 5.2 Metrics

We evaluate agent performance with several metrics. These are based on guidelines by Anderson et al. [2] and followed in other work [25, 39, 29, 30, 16, 14], but extended to account for multiple object goals, as well as to give partial credit for navigating to a subset of goals.

SUCCESS: Binary indicator of episode success. Episode is successful if the agent navigates to all goals in correct order, calling FOUND at each goal, within the maximum number of steps allowed. An incorrectly called FOUND terminates the episode immediately.

PROGRESS: The fraction of object goals that are successfully FOUND. Equal to success for 1-ON.

SPL: Extension of 'Success weighted by Path Length' [2] to the multiON task. Concretely, SPL $= s \cdot d/\max(p, d)$ where $s$ is the binary success indicator, $p$ is the total distance traveled by the agent, and $d = \sum_{i=1}^{n} d_{i-1,i}$ is the total geodesic shortest path distance from the agent's starting point through each goal position in order with $d_{i-1,i}$ indicating geodesic distance from goal $i-1$ to goal $i$, and goal 0 being the starting point.

PPL: A version of SPL based on progress: PPL $= \bar{s} \cdot \bar{d}/\max(p, \bar{d})$ where $\bar{s}$ is the PROGRESS value, $\bar{d} = \sum_{i=1}^{l} d_{i-1,i}$ with $l$ being the number of objects found, and $p$ and $d_{i-1,i}$ are defined as before. The overall path length is weighted by progress, rather than weighing individual subgoal discoveries by their respective path lengths so as to not assign disproportionately high weights to shorter goal-to-goal trajectories within the episode. PPL is equal to SPL for 1-ON.

## 5.3 Example episodes

Figure 3 shows example episodes visualizing the performance of different agent models on val set scenes. NoMap(RNN) tends to wander and get low PPL even when it is able to reach the goals. When FOUND is called not in the vicinity of the current goal, the episode is terminated resulting in failure. This happens in the top row for several agent models, as well as ProjNeuralMap in the bottom row.

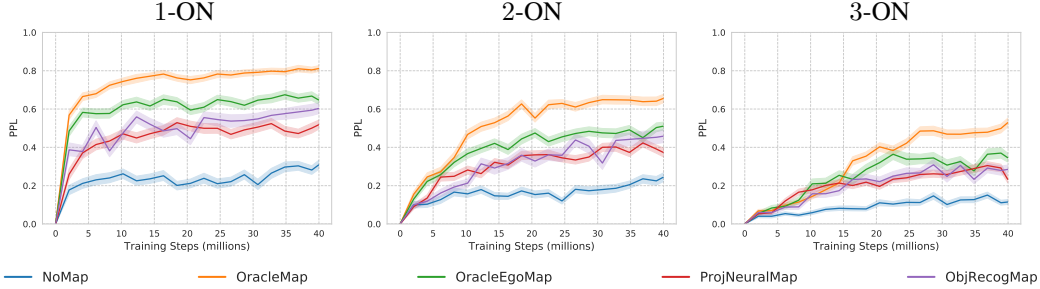

Figure 4: PPL of agents during training, evaluated on validation set with $95\%$ confidence intervals in shading. Overall performance decreases for all models as task complexity increases. `OracleMap` trains the fastest and reaches the best overall performance. `OracleEgoMap` and `ObjRecogMap` follow closely, while the `ProjNeuralMap` and `NoMap(RNN)` agents perform the worst.

## 5.4 Quantitative evaluation

We train all agents for $40$ million steps on 1-ON, 2-ON and 3-ON tasks. During training we measure performance on $1,000$ validation scene episodes. The maximum number of steps is $2,500$, which we found to be sufficiently large for all $m$-ON tasks in this paper. Figure 4 plots the PPL metric for all agent models. We pick the best performing checkpoint for each model and evaluate on $1,000$ episodes in $18$ unseen test scenes. Table 2 summarizes all evaluation metrics averaged across test episodes (including for ablations of the `OracleMap` agent using only occupancy or object category information). As expected, the `OracleMap` agents have the highest performance across the board, with significant gains over the NoMap baselines (`RNN`, `SMT`, `FRMQN`) indicating the value of explicit map information. Of the `NoMap` variants, `FRMQN` performed the best, surpassing the newer transformer-inspired model `SMT` (potentially due to the need for more training). Object category information in the map accounts for most of the gain. The `OracleEgoMap` agent follows, but `ObjRecogMap` is surprisingly close, followed by `ProjNeuralMap`.

*How hard are the multiON tasks?* Even the oracle agents drop to $\leq 50\%$ PPL on 3-ON. Our `Rand+OracleFound` agent is similar to `Random` in Fang et al. [21]. At 500 steps Fang et al. [21]'s `Random` finds $36.3\%$ of object classes, whereas `Rand+OracleFound` finds $9\%$, $5\%$ and $3\%$ in 1-ON, 2-ON, 3-ON respectively (note that Table 2 reports metrics at $2,500$ max steps). This indicates the $m$-ON tasks are quite challenging. In particular, objects need to be found in order, so we see a clear trend where $m$-ON is harder with more goals $m$. The multiON task is challenging and well-suited to investigating the usefulness of map information across a spectrum of difficulties. The performance drop between 1-ON and 3-ON is dramatic. We might have expected agents that remember past goal observations to have a higher rate of success when returning to a goal later. In the supplement we carry out an analysis verifying that some agents can remember the location of objects seen during the course of the episode. Despite this, we observe a worse than purely exponential drop in performance. With exponential decay, we would expect `OracleMap` SUCCESS rates to be $0.94$, $0.94^2 = 0.88$ and $0.94^3 = 0.83$ as $m$ is increased from 1 to 3. The observed SUCCESS rates are $0.94$, $0.79$ and $0.62$, indicating that there is room for improvement.

*Are spatial maps useful?* Yes. Having ground truth maps helps significantly (compare `OracleMap` and `OracleEgoMap` with `NoMap(RNN)`). `OracleMap` is almost perfect ($94\%$ PROGRESS/SUCCESS) on 1-ON, but does not always pick the shortest path (SPL/PPL of $77\%$). However, performance drops for 2-ON and 3-ON. Still, maps help more for 2-ON and 3-ON than for 1-ON indicating the value of spatial memory for longer-horizon tasks.

*What information in the spatial maps is useful?* While it may be obvious that maps help, we see that object category information in the map indicating the goal location is much more useful than occupancy information (see drop from `OracleMap (Obj)` to `OracleMap (Occ)`). Our hypothesis is that occupancy mainly helps to avoid collisions and this can be achieved using the depth sensor, since we have perfect depth and localization (i.e. no actuation or sensor noise). Note that the `Obj` ablation is higher than `Occ+Obj`. We suspect this is due to `Occ+Obj` being harder to train to leverage the added information channel, as `Occ+Obj` did not fully converge within $40M$ training steps.

Table 2: Agent performance on 1-ON, 2-ON and 3-ON test set (maximum 2,500 steps). The multiON task is challenging with `Rand+OracleFound` achieving $26\%$ success (SPL $8\%$) for 1-ON, and `Rand` failing completely. Performance decreases for all agents as we add more objects. Overall, maps help considerably, with the ability to represent goal objects in the map being particularly valuable (compare `OracleMap (Obj)` and `OracleMap (Occ)` as well as `ObjRecogMap` and `ProjNeuralMap`).

| | | Success (%) | | | Progress (%) | | | SPL (%) | | | PPL (%) | | |
|---|---|---|---|---|---|---|---|---|---|---|---|---|---|
| | | 1-ON | 2-ON | 3-ON | 1-ON | 2-ON | 3-ON | 1-ON | 2-ON | 3-ON | 1-ON | 2-ON | 3-ON |
| | Rand | 0 | 0 | 0 | 0 | 0 | 0 | 0 | 0 | 0 | 0 | 0 | 0 |
| | Rand+OracleFound | 26 | 8 | 2 | 26 | 16 | 12 | 8 | 2 | 1 | 8 | 5 | 4 |
| NoMap | NoMap(RNN) | 62 | 24 | 10 | 62 | 39 | 24 | 35 | 13 | 4 | 35 | 21 | 14 |
| | FRMQN [34] | 62 | 29 | 13 | 62 | 42 | 29 | 50 | 24 | 11 | 50 | 33 | 24 |
| | SMT [21] | 63 | 28 | 9 | 63 | 44 | 22 | 48 | 26 | 7 | 48 | 36 | 18 |
| Oracle | OracleMap (Occ+Obj) | 94 | 74 | 48 | 94 | 79 | 62 | 77 | 59 | 38 | 77 | 63 | 49 |
| | OracleMap (Occ) | 66 | 34 | 16 | 66 | 47 | 36 | 48 | 25 | 12 | 48 | 35 | 27 |
| | OracleMap (Obj) | 94 | 82 | 59 | 94 | 86 | 70 | 79 | 65 | 42 | 79 | 67 | 50 |
| | OracleEgoMap (Occ+Obj) | 83 | 64 | 37 | 83 | 71 | 54 | 65 | 49 | 25 | 65 | 54 | 36 |
| Learned | EgoMap [7] | 69 | 46 | 26 | 69 | 59 | 44 | 49 | 31 | **18** | 49 | 42 | 30 |
| | ProjNeuralMap | 70 | 45 | **27** | 70 | 57 | **46** | 51 | 30 | **18** | 51 | 39 | **31** |
| | ObjRecogMap | **79** | **51** | 22 | **79** | **62** | 40 | **56** | **38** | 17 | **56** | **45** | 30 |

*What kind of learned map is useful?* Surprisingly, `ObjRecogMap` which simply learns to project predicted object categories into the map is quite competitive and outperforms the `ProjNeuralMap` and `EgoMap` agents for 1-ON and 2-ON based on the MapNet [27] projection architecture. For 3-ON, the `ProjNeuralMap` and `EgoMap` agents are able to outperform `ObjRecogMap`. Overall, the performance of `ProjNeuralMap` and `EgoMap` are very similar, suggesting limited contribution of using attention to read from the semantic map. This indicates that simple recognition of the goal from visual features and direct integration into a map memory can be more effective than accumulating image features into a map. Still, performance is not as good as `OracleEgoMap` indicating that there is more work to be done on learned map representations to match oracle egocentric mapping.

*What's next?* Even with `OracleMap` success drops to only $59\%$ ($42\%$ SPL) for 3-ON, despite the apparent triviality of navigating to three distinct objects. A key question for future work is how to design architectures that avoid a dramatic performance drop with increasing numbers of goal objects, which is a clear sign of non-robustness in embodied agents. The drop between `OracleMap` and `OracleEgoMap` is likely attributed to partial observability of the environment under egocentric embodiment constraints and offers a more conservative 'high bar' for future work. The subsequent drop from `OracleEgoMap` to both `ProjNeuralMap` and `ObjRecogMap` indicates that further study of approaches for aggregating egocentric information and integrating it into a map would be fruitful.

In this paper, we investigated a representative set of prior work on agent map representations and compared their performance on the MultiON task. We have assumed perfect localization in our experiments, which is usually not a practical assumption. An interesting direction would be to explore how neural inspired grid-cell architectures such as in Banino et al. [5] that predict the location and heading can be used to localize the agent. We have also focused on the use of grid-based maps. Other types of maps such as topological maps would be interesting to investigate and contrast.

# 6 Conclusion

We introduced multiON, a task framework allowing for systematic analysis of embodied AI navigation agents utilizing semantic map representations. Our experiments with several agent models show that semantic maps are indeed highly useful for navigation, with a relatively naïve integration of semantic information into map memory providing high gains against more complex learned map representations. However, overall agent performance degrades dramatically with modest task difficulty increases even for the best performing agents. Our experiments suggest several promising directions for future work, including improved learned modules for integrating egocentric information into map representations. We hope that the multiON framework provides a flexible benchmark for systematic study of spatial memory and mapping mechanisms in embodied navigation agents.

## Acknowledgments

We thank the anonymous reviewers for their helpful suggestions. Unnat Jain thanks Alexander Schwing and Svetlana Lazebnik for their support. Angel X. Chang is supported by the Canada CIFAR AI Chair program. Manolis Savva is supported by an NSERC Discovery Grant. This research was enabled in part by support provided by WestGrid (https://www.westgrid.ca/) and Compute Canada (www.computecanada.ca).

## Broader Impact

This work is a step toward enabling robots that can navigate and operate in real world environments. We focus on the study of what kind of maps might be useful to such robotic agents, and how to benchmark their performance. In the future, robust robotic agents may be able to assist the elderly (bringing them their medication), deliver things to hotel rooms, help in moving items in warehouses, and generally serve broader society in a variety of roles. On the other hand, success along these directions may cause displacement of workers from related occupations and economic difficulties for large segments of the population currently employed in a variety of sectors that are amenable to automation. Moreover, deployment of imperfect robotic agents, that are not guaranteed to be failure-free may cause injuries or damages. We believe that developing and studying the behavior of such systems in simulation first, and then in controlled real environments is paramount for minimizing these risks, before they are deployed in the real world.

## Footnotes

*denotes equal contribution

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
