[Supplementary Material]

# A  Supplementary material

In this supplemental document, we first provide a summary of the notation used in the main paper and here (Appendix A.1). Then, we describe the implementation details of agent models used for our experiments (Appendix A.2). In Appendix A.3 we provide statistics for the episodes used in our experiments. Finally, Appendix A.4 contains additional experiments and analysis.

## A.1  Notation

Table 3 provides a summary of definitions for important symbols used in the main paper and in this supplemental document.

Table 3: Summary of notation used. Subscript $t$ denotes the corresponding notation at time step $t$

| Notation | Description | Notation | Description |
|---|---|---|---|
| $m$-ON | Episode with $m$ ordered object goals | $v_t$ | concat$(v_i, v_m, v_g, v_a)$ |
| $p_t$ | Agent's position and orientation | $s_t$ | Final state representation |
| $o_t$ | Egocentric RGBD sensor images | $\theta$ | Parameters of end-to-end trainable model |
| $c_t$ | Egocentric RGB sensor image | $\pi_\theta(\cdot \mid s_t)$ | Actor policy given state $s_t$ |
| $d_t$ | Egocentric Depth sensor image | $V(s_t)$ | Approximate value function |
| $g_t$ | Current goal object one-hot vector | $M_{GT}$ | Oracle Map |
| $a_t$ | Action taken by the agent | $r_t$ | Reward |
| $m_t$ | Egocentric Map | $r_{\text{subgoal}}$ | Subgoal discovery reward |
| $M_t$ | Global Map | $r_{\text{closer}}$ | Moving closer to subgoal reward |
| $i_t$ | Image Features | $r_{\text{time-penalty}}$ | Time penalty reward |
| $v_i$ | Transformed Image features after passing image through a CNN and a linear layer | $\alpha$ | Negative slack reward |
| $v_i$ | Embedding of $m_t$ after passing it through a CNN and a linear layer | $d_{i-1,i}$ | Geodesic distance of the shortest path between goal $i-1$ and $i$ |
| $v_g$ | Embedding of one-hot goal vector $g_t$ | $s$ | Binary success indicator |
| $v_a$ | Embedding of previous action $a_{t-1}$ | $\bar{s}$ | PROGRESS |

## A.2  Agent model details

We describe the details of the `OracleEgoMap`, `ObjRecogMap`, and `ProjNeuralMap` agent models. For each map-based model, the agent has access at time $t$ to a global map $M_t$ which is a $300 \times 300$ grid, with each cell corresponding to a 0.8m $\times$ 0.8m square in the environment. Depending on the model, $M_t$ is revealed or built up over time. The details of what goes into each map cell and how the information is built up over time are described below.

`OracleEgoMap`: At each time step, the agent has access to a partially revealed map $M_t$ that is derived from the oracle map. Each cell has two channels: i) an occupancy channel; and ii) an object category channel. The occupancy channel stores a 16-dimensional embedding learned from whether the cell is navigable, non-navigable or *undiscovered*. The object category channel stores a 16-dimensional embedding of the the category of the object occupying that cell learned from a $k+1$ dimensional one-hot vector ($k$ dimensions for the goal categories and $(k+1)^{th}$ dimension for 'no goal'.) At the start of the episode, all the cells in the occupancy channel are 'undiscovered'. A cell is *discovered* once the agent sees (the region of the environment corresponding to) it and it remains discovered through the rest of the episode. At each time step, the agent sees only those locations in the environment that are: i) within a 79° field-of-view in front of it; ii) less than 5m away from the agent; and iii) with no obstacle between the agent and that location. All cells in the object category channel are initialized with the $(k+1)^{th}$ category embedding. When an object is discovered, its category embedding is stored in the corresponding cell in the object category channel.

`ObjRecogMap`: The map $M_t$ used in `ObjRecogMap` is similar to that of `OracleEgoMap` but instead of it being 'revealed' from the oracle map, it is predicted based on egocentric views. In addition, each cell has only the object category channel. As in `OracleEgoMap`, each cell stores one of the $k+1$ 16-dimensional embeddings ($k$ embeddings for the goal categories and $(k+1)^{th}$ for 'no goal'.)

At each time step, the current RGB frame is passed through a $(k + 1)$—classification network to predict the category of the goal in view (if no goal is in view, $(k + 1)$ is used as the prediction target). At training time, the network is supervised using the ground truth object category information to predict which goal object is in view. An object is considered to be in the agent's view if it falls in a grid cell that satisfies the three conditions described in above for `OracleEgoMap`. We use categorical cross-entropy loss to supervise the network. If more than one object is in the agent's current view, the agent is trained to predict the one that is closer to it. Initially, all the cells in the map store the $(k + 1)^{th}$ category embedding (corresponding to 'no goal'.) If the agent predicts $(k + 1)^{th}$ category at a time step, the map is not updated. If the agent predicts a category $l < k$, it stores the encoding for the $l^{th}$ category at the cell containing its current position.

`ProjNeuralMap`: As before, the agent has access to a partially built map $M_t$. The agent builds up an egocentric map $m_t$ of size $7 \times 13$. Note that this covers an area of $5.6m$ in front and to both sides of the agent. Points that are farther than $5.6m$ from the agent are not projected. Here also, the agent has a $79°$ field-of-view. We project the image feature $i_t(i, j, \cdot)$ on the ground using the depth frame $d_t$. To do this, we downsample $d_t$ so it matches the image feature dimensions and use the downsampled depth for projection. We observed that the alternative of upsampling and interpolating image features leads to reduced performance. The agent is at the mid bottom edge of this egocentric map. $R(m_t, M_t|p_t)$ registers the egocentric map $m_t$ into $M_t$. The integration into $M_t$ uses element-wise max-pooling.

## A.3 Episode statistics

Figure 5 plots the distribution of episodes from the train/val/test splits across total episode geodesic distance along the oracle shortest path. Note that overall, the distribution tends to longer episodes as we go from 1-ON to 2-ON and 3-ON. There are small variations between the train/val/test splits for each $m$-ON, but the distributions are generally in correspondence.

Figure 5: Plots showing the distribution of train, val and test split episodes over the total geodesic distance for the oracle shortest path of the episode. The horizontal axis denotes the geodesic distance and the vertical axis denotes the fraction of total episodes in the corresponding histogram bin. The first row corresponds to the train split, the second row corresponds to the val split, and the third row corresponds to the test split. Each scene has 50,000 episodes in the train split and 12,500 episodes in the val and the test splits.

## A.4 Additional experiments and analysis

Table 4: Success rate (%) for finding the $k^{th}$ subgoal in the 3-ON task. The 'seen' column has the accuracy with which the $k^{th}$ subgoal is found if it was *seen* before $(k-1)^{th}$ subgoal was discovered. In contrast the 'not seen' column represents episodes where the goal was not previously seen. The 'improvement' column reports the difference. As expected the `NoMap(RNN)` and `OracleMap` do not exhibit improvements. In contrast, storing previously observed goal information helps `OracleEgoMap`, `ProjNeuralMap` and `ObjRecogMap` significantly, with `ObjRecogMap` showing the largest gains.

| | Second goal ($k = 2$) | | | Third goal ($k = 3$) | | |
|---|---|---|---|---|---|---|
| | seen | not seen | improvement | seen | not seen | improvement |
| `NoMap(RNN)` | 53 | 51 | +2 | 47 | 46 | +1 |
| `OracleMap` | **82** | **80** | +2 | **80** | **79** | +1 |
| `OracleEgoMap` | 80 | 66 | +14 | 76 | 59 | +17 |
| `ProjNeuralMap` | 74 | 54 | +**20** | 68 | 44 | +24 |
| `ObjRecogMap` | 72 | 53 | +19 | 69 | 44 | +**25** |

**Backtracking analysis.** We compare the success rate for finding the $k^{th}$ ($k = 2$, $k = 3$) goal in two types of scenarios: i) the $k^{th}$ goal was *seen* before the $(k-1)^{th}$ subgoal was FOUND; and ii) the $k^{th}$ subgoal was *not seen* before the $(k-1)^{th}$ subgoal was FOUND. The results for 3-ON episodes are summarized in Table 4. A goal object is *seen* if it satisfies the three conditions mentioned in Appendix A.2. This analysis quantifies the ability of the agent to *remember* the location of an object that was previously seen. If an agent sees the second goal while it is looking for the first goal, it would benefit from this information at the time of searching for the second goal. Understandably, this has little effect for the `NoMap(RNN)` agent since it has no way of storing the location of previously seen goals, beyond implicit encoding in the GRU module of the agent architecture. `OracleMap` too does not benefit much since it already has the full oracle map of the environment. On the other hand, `OracleEgoMap` and `ObjRecogMap` exhibit a significant jump in success rate when the goal was previously observed. Notably, `ProjNeuralMap`, which stores information about what was observed but does not explicitly convert it to goal category, also shows a small gain for previously observed goals but less than that of `OracleEgoMap` and `ObjRecogMap`. These results further demonstrate the value of map memory and the importance of the kind of information stored in the map for the multiON task.

Table 5: Evaluation of agents trained on $m$-ON (in rows) test episodes of $n$-ON (in columns). We report the PROGRESS metric averaged across all test set episodes. Off the diagonal for each agent shows generalization capability to multiON episodes with a different number of objects than in training. The top row triplet reports absolute values, while the bottom one reports change in PROGRESS relative to the diagonal (evaluation on trained task). As expected, performance drops for all agents in all generalization tests, with the drop being greater for testing on episodes with more objects. Agents trained on 1-ON have the lowest generalization capability, while 2-ON and 3-ON agents exhibit smaller but still significant drops in performance.

| | NoMap(RNN) | | | OracleMap | | | OracleEgoMap | | | ProjNeuralMap | | | ObjRecogMap | | |
|---|---|---|---|---|---|---|---|---|---|---|---|---|---|---|---|
| | 1-ON | 2-ON | 3-ON | 1-ON | 2-ON | 3-ON | 1-ON | 2-ON | 3-ON | 1-ON | 2-ON | 3-ON | 1-ON | 2-ON | 3-ON |
| 1-ON | 62 | 23 | 11 | 94 | 27 | 12 | 83 | 21 | 9 | 65 | 25 | 12 | 79 | 21 | 12 |
| 2-ON | 56 | 39 | 22 | 89 | 79 | 46 | 8 | 71 | 43 | 71 | 57 | 44 | 77 | 62 | 38 |
| 3-ON | 53 | 38 | 24 | 84 | 76 | 62 | 77 | 7 | 54 | 67 | 56 | 46 | 64 | 55 | 40 |
| 1-ON | 0 | -16 | -13 | 0 | -52 | -5 | 0 | -5 | -54 | 0 | -32 | -32 | 0 | -41 | -28 |
| 2-ON | -6 | 0 | -2 | -5 | 0 | -16 | -3 | 0 | -11 | 6 | 0 | -2 | -2 | 0 | -2 |
| 3-ON | -9 | -1 | 0 | -1 | -3 | 0 | -6 | -1 | 0 | 2 | -1 | 0 | -15 | -7 | 0 |

**Testing agent generalization.** We report task generalization performance in Table 5 where agent models trained on an $k$-ON task are evaluated on $l$-ON task where $k \neq l$. Models trained on 1-ON fail to generalize to more complex multiON tasks for both agents with and without map memory. This is likely caused by no prior training for 'continuing' the task past the first found goal (i.e. the agents have not been trained to withhold calling FOUND until they navigate to another goal after the first one). In contrast, the agents trained on 2-ON and 3-ON are able to generalize to some extent, especially for the case of going from a more complex task to a less complex one. This is exhibited by smaller performance drops in the 'lower triangle' than the 'upper triangle' of the generalization table.

**Analysis of agent performance vs episode length.** We analyze agent performance based on the episode complexity, as measured by the geodesic distance of the oracle shortest path between the start position and all the goals in the episode. Figure 6 shows the average PPL for each agent across a range of total episode geodesic distance along the oracle path. Note that the `OracleMap` agent is able to have relatively high PPL regardless of the episode geodesic distance (with the overall PPL dropping somewhat from 1-ON to 3-ON). `NoMap(RNN)` has relatively low PPL and often fails to reach any goals for harder episodes (i.e. episodes with higher geodesic distance). For both `OracleEgoMap` and `ObjRecogMap`, we see that the PPL decreases as we go from easier episodes to harder episodes, indicating the challenge in achieving agent robustness.

**Agent performance during training.** In Figure 7 we plot the SPL, SUCCESS, and PROGRESS metrics for the different agents on the validation set (see main paper Figure 4 for the PPL). All the metrics follow the same general trends as observed in the main paper.

**Additional episode visualizations.** Figure 8 shows additional visualizations of test set episodes. We see that the episodes span a range of environments, with fairly complex paths between goals that frequently require some degree of backtracking. As we saw in the analysis of agent performance when goals are previously seen vs not, such episodes requiring backtracking help us to benchmark the ability of the agents to store and use information on previously seen goals.

Figure 6: Plots of average PPL against geodesic distance along oracle path (i.e. horizontal axis bins episodes into ranges of total episode geodesic distance along the oracle path, and the vertical axis shows average PPL for all episodes in the corresponding bin). There is a general trend of decreasing average PPL with increasing episode distance for all the models (though `OracleMap` shows a relatively small drop in performance). This indicates the escalating difficulty of the multiON task as the length of the episodes is increased.

Figure 7: Evaluation metrics of agents during training, evaluated on validation set with $95\%$ CI indicated by shading. Overall performance decreases across all models as task complexity increases. The `OracleMap` agent trains the fastest and reaches the best overall performance. `OracleEgoMap` and `ObjRecogMap` follow closely, while the `ProjNeuralMap` and `NoMap(RNN)` agents perform the worst.

| NoMap(RNN) | OracleMap | OracleEgoMap | ProjNeuralMap | ObjRecogMap |
|---|---|---|---|---|

Goal order: 1●, 2●, 3●

| P=0, PPL=0 | P=1, PPL=0.95 | P=1, PPL=0.54 | P=0, PPL=0 | P=1, PPL=0.66 |

Goal order: 1●, 2●, 3●

| P=0.33, PPL=0.06 | P=1, PPL=0.53 | P=1, PPL=0.87 | P=0.33, PPL=0.07 | P=0, PPL=0 |

Goal order: 1○, 2●, 3●

| P=0, PPL=0 | P=1, PPL=0.71 | P=1, PPL=0.49 | P=0, PPL=0 | P=0.33, PPL=0.28 |

Goal order: 1●, 2●, 3●

| P=0.33, PPL=0.13 | P=0.33, PPL=0.30 | P=1, PPL=0.44 | P=0, PPL=0 | P=0.66, PPL=0.51 |

Goal order: 1●, 2●, 3●

| P=0, PPL=0 | P=1, PPL=0.91 | P=1, PPL=0.53 | P=0, PPL=0 | P=0.66, PPL=0.53 |

Goal order: 1●, 2●, 3○

| P=0.66, PPL=0.48 | P=1, PPL=0.97 | P=0.66, PPL=0.17 | P=0, PPL=0 | P=1, PPL=0.84 |

Figure 8: Additional example episodes for different agents. Agent path and shortest path in orange and green, with the start shown by ☐ (orange square).

Table 6: Test set performance for ablations of the `OracleMap` model. `DynamicOracleMap` is dynamically updated to only indicate the current goal object instead of all goals. `OnlyOracleMap` shows results with only map information and no RGBD sensors. The `OracleMapHiddenObjects` is the same as `OracleMap` except the goal objects are not inserted into the environment so the agent must navigate to the target locations based on the map information alone.

| | SUCCESS (%) | | | SPL (%) | | |
| --- | --- | --- | --- | --- | --- | --- |
| | 1-ON | 2-ON | 3-ON | 1-ON | 2-ON | 3-ON |
| `OracleMap` | 94 | 74 | 48 | 77 | 59 | 38 |
| `DynamicOracleMap` | 94 | 85 | 81 | 74 | 71 | 66 |
| `OnlyOracleMap` | 54 | 9 | 0 | 42 | 7 | 0 |
| `OracleMapHiddenObjects` | 91 | 72 | 40 | 75 | 57 | 33 |

**Oracle map ablations.** We perform a series of ablation experiments to investigate how information provided in the oracle map and its connection with other sensory input during the task can determine agent performance. The results of these experiments are in Table 6 and are summarized below.

- *Dynamically updated oracle map.* Instead of storing the embedding of all target objects in the oracle map at the beginning of the episode, we store the embedding of only the current target object and dynamically update the map when the agent finds a target object. This helps simplify training since the agent does not have to distinguish between the embeddings of various objects stored in the map (only the current object embedding is stored). This effectively breaks down an $m$-ON task into $m$ ObjectNav (1-ON) tasks. As expected, the SUCCESS rate in an $m$-ON task is approximately $s^m$, where $s$ is the SUCCESS rate of 1-ON. Note that this is significantly better than the SUCCESS rate in the original setup where all the object embeddings are stored in the oracle map at the start of the episode.
- *Map-only baseline.* We experiment with agents that do not have RGBD sensors and must only use the oracle map to navigate to their target locations in an $m$-ON task (`OnlyOracleMap`). Although the agent has access to the locations of all the target objects in the environment (in the form of embeddings stored in the map), it fails to perform as well as the agent using RGBD sensors. The SUCCESS rate drops rapidly from 1-ON to 2-ON to 3-ON. This could be due to two reasons: 1) the agent is unable to recognize the target object and call FOUND when it is near the object, or 2) the agent is unable to navigate (avoid obstacles, walk through hallways etc.) effectively through the environment. Further experiments and qualitative analysis of rollouts from validation episodes reveal that the decreased performance is primarily due to the inability of the agent to navigate effectively through the scenes.
- *Hidden goal objects.* In this variant (`OracleMapHiddenObjects`), the agent is provided an oracle map and has access to RGBD sensory information but the objects are not actually inserted in the environment. The agent must therefore infer the location of the target objects from the oracle map. The RGBD sensors are expected to help learn basic navigation skills like obstacle avoidance. As is evident from Table 6, the model performs just as well as the one where objects are inserted in the environment. This also shows that the fall in performance in the `OnlyOracleMap` baseline can largely be attributed to a lack of basic navigation skills rather than an inability to call FOUND action at the right time.
- *Varying map resolution.* We experimented with different map resolutions and sizes of oracle maps during validation. Final results are given with the resolution settings that we found to perform best. The resolution of the map refers to the physical length of a single grid cell in the environment. The size of the map refers to the the area of the global map that is cropped (and rotated) for generating the egocentric map $m_t$. Similarly to Chen et al. [17], Chaplot et al. [14], we experimented with a variant where we stack both high and low resolution maps together. Here, the map consists of 4 channels, 2 each for the high and low resolution maps. The two maps correspond to different area sizes in the environment and therefore provide the agent with more local or more global views of the scene. We did not find any improvements in agent performance over the base `OracleMap` model.

Table 7: Test set performance for ablations of the `OracleEgoMap` model. `DynamicOracleEgoMap` is dynamically updated to only indicate the current goal object (if it has been seen yet) instead of all goals. `DynamicObjRecogMap` is dynamically updated to only indicate the current goal object (it is has been seen and identified).

| | SUCCESS (%) | | | SPL (%) | | |
|---|---|---|---|---|---|---|
| | 1-ON | 2-ON | 3-ON | 1-ON | 2-ON | 3-ON |
| `DynamicOracleEgoMap` | 83 | 62 | 41 | 65 | 46 | 25 |
| `DynamicObjRecogMap` | 79 | 52 | 20 | 56 | 35 | 16 |

Table 8: Test set performance of `NoMap(RNN)` baseline when varying color and shapes.

| | SUCCESS (%) | | | PROGRESS (%) | | | SPL (%) | | | PPL (%) | | |
|---|---|---|---|---|---|---|---|---|---|---|---|---|
| | 1-ON | 2-ON | 3-ON | 1-ON | 2-ON | 3-ON | 1-ON | 2-ON | 3-ON | 1-ON | 2-ON | 3-ON |
| `NoMap(RNN) (color)` | 62 | 24 | 10 | 62 | 39 | 24 | 35 | 13 | 4 | 35 | 21 | 14 |
| `NoMap(RNN) (shape)` | 73 | 1 | 0 | 73 | 9 | 5 | 55 | 1 | 0 | 55 | 6 | 4 |

**Dynamically updating maps**. Similar to ablations with dynamically updated oracle maps, we experiment with variants of other models in which only the embedding of the current target object is stored in the object channel of the map. This is not possible with `ProjNeuralMap` since we do not store object embeddings. In the `DynamicOracleEgoMap`, if the current goal has been discovered its embedding is stored in the map. If not, the object channel of the map does not store anything. In the `DynamicObjRecogMap`, if the current goal has been discovered (and correctly identified through object classification), its embedding is stored in the map. Otherwise, the object channel does not store any information. Table 7 summarizes the performance metrics for these variants. We see that these variants do not perform better than the base models in which the embeddings of all target objects discovered so far are stored in the map.

**Maximum episode length determination.** The maximum allowed episode length for all experiments is 2,500. This limit is large enough that it does not affect the success rate in an episode. To ensure that this threshold does not influence our results, we evaluated all agents with the time limit set proportional to $m$ (2,500 for 3-ON, $\frac{2}{3} \times 2,500$ for 2-ON and $\frac{1}{3} \times 2,500$ for 1-ON). This proportional thresholding has negligible impact on the performance metrics. Further, the mean and median episode lengths are 276.2 and 151 steps respectively for 3-ON experiments, which shows that most episodes terminate by calling a wrong FOUND action rather than by reaching the maximum time limit.

**Varying target object shapes.** We vary the shapes of the target objects, by choosing from 8 different objects: cone, cube, cylinder, frustum, joined inverted frustums, sphere, tetrahedron and torus. All these objects have a horizontal length/diameter of $0.4m$. They have the same red color and are all placed on the top of a thin cylindrical support. Table 8 compares scenarios with these varying shape targets to the old scenarios where shape was constant and color was varying instead. 1-ON performs better in this new setting but 2-ON and 3-ON perform worse. It is perhaps easier for the agent to identify goals if they are close to camera height as opposed to identifying goals that extend across the height of the agent. So this new setting favors the agents in identifying goal objects, but it is more difficult to differentiate between different target objects.

**Effect of having a *hard* FOUND action.** An episode of $m$-ON can terminate either by calling a wrong FOUND action or by reaching the time limit (2,500 for all experiments in this paper). Calling a FOUND action when not within the threshold distance of the current target object leads to a 'wrong' FOUND action. To understand the effect of having a *hard* FOUND action, where a single wrong FOUND terminates the episode immediately, we allow the agent to call a fixed number of wrong FOUND actions during the episode. We found that allowing even a single wrong FOUND action leads to a significant increase in performance metrics. This suggests that many episodes terminate due to calling FOUND action at the wrong time and fixing this inadequacy could improve $m$-ON performance significantly. Table 9 summarizes the results of performance metrics against the number of wrong

Table 9: Test set performance of `OracleMap` on the 3-ON task while allowing a fixed number of wrong FOUND actions during the episode. First column (FOUND budget) lists the number of wrong FOUND actions allowed while the other columns list the corresponding performance metrics.

| FOUND budget | PROGRESS (%) | SUCCESS (%) | PPL (%) | SPL (%) |
|---|---|---|---|---|
| 0 | 62 | 48 | 49 | 38 |
| 1 | 76 | 66 | 58 | 50 |
| 2 | 79 | 70 | 58 | 50 |
| 3 | 82 | 74 | 59 | 54 |
| 5 | 85 | 78 | 60 | 55 |
| 10 | 87 | 82 | 60 | 55 |
| 15 | 88 | 82 | 60 | 56 |
| 20 | 90 | 85 | 60 | 57 |
| 50 | 90 | 85 | 60 | 57 |
| Oracle Found | 93 | 89 | 65 | 62 |

FOUND actions allowed in `OracleMap` model on the 3-ON task. Note that these evaluations were performed on agents trained in the usual way, where a single wrong FOUND action terminates the episode. The last row of the table (Oracle Found) quotes the performance metrics when an 'oracle' issues a FOUND action when the agent is within the threshold distance of the current target object. For this, the action is sampled from the three action probabilities corresponding to {FORWARD, TURN-LEFT, TURN-RIGHT} and FOUND action is called automatically when the agent is near the current target goal.