[Reviews · NeurIPS 2020]

Review 1

Summary and Contributions: The paper describes a new benchmark navigation environment designed specifically to test out and compare the capabilities of various memory map architectures. The proposed "MultiON" task requires an agent to navigate and acquire several randomly located objects in Matterport3D scenes in a specific order. The paper re-implements several baselines models, including a simple LSTM baseline, a MapNet-style egocentric map, a novel baseline of an allocentric map populated with seen object categories, and various oracle map baselines with access to privileged information such as occlusion and object location data. Results demonstrate (1) that object category data seems more important than what might have previously been assumed and (2) that current memory map architectures have a large gap from oracle baselines. Overall, MultiON creates a unified platform for evaluating memory map architecture and motivates further research in this direction by demonstrating a large gap in performance left to be closed by future work.

Strengths: I believe the paper has strong significance to the field of navigation architectures. It firstly presents a series of benchmarks and evaluates a comprehensive set of baselines on these benchmarks. Secondly, the results demonstrate that (1) the environments are difficult and will require new methods to be developed (i.e. even oracle baselines do not perform close to optimally) and (2) that even between existing models and oracle baselines, there is a large gap to be filled. Therefore the work both establishes a very clear need for improvement in current state-of-the-models and provides a challenging setting (and baseline results) in which future work can evaluate and compare new models. The benchmark also fills a unique niche within released benchmarks by focusing on map architectures. Beyond the establishment of a new benchmark suite, the paper's empirical results provide some interesting novel insight into map architectures. For example, that object recognition seems to be a strong component of improved performance demonstrates that purely RL-driven feature learning might not be the optimal choice. Furthermore it shows that the projection step designed in MapNet does not always provide a benefit, with that model slightly improving over an LSTM baseline. This provides some evidence that moving towards a semantic map memory that obeys stricter geometric relationships between feature positions does not consistently result in better performance. The empirical results are comprehensive and follow previous best practices of the field. The analysis in the supplementary material is very interesting, and provides insight into why certain models perform better than others.

Weaknesses: The paper and results currently all rely on semantic maps as the main memory mechanism driving navigation, but there are other approaches such as the Scene Memory Transformer SMT) referenced in the paper. Although I think it is fine for this work to focus results on map architectures, it would be interesting to have at least one result comparing how an alternative method like the SMT would perform. Especially since the SMT contained an environment similar to MultiON on which it was much more successful than baselines. I have a question on the number of seeds run per model, as I could not find any reference to this number. If there is only 1 seed run, I think that would be a weakness of the paper as RL algorithms can sometimes have a significant difference in final performance between initial seeds.

Correctness: I did not see any incorrect claims or methods presented in the paper. The empirical methodology seems correct and follows guidelines established in previous work on navigation.

Clarity: I enjoyed reading the paper as it is very well-written and easy to read. There are sufficient details on all architectures and tasks to easily understand the results presented.

Relation to Prior Work: While there are other navigation benchmarks, the paper clearly distinguishes itself from these. The paper also clearly emphasizes why this new benchmark would be valuable to the field. Beyond previous benchmarks, the related work contains a broad overview of previous papers on navigation architectures.

Reproducibility: Yes

Additional Feedback: Post-Rebuttal: After reading the other reviews and rebuttal, I remain positive about this paper and keep my vote for acceptance.


Review 2

Summary and Contributions: The paper provides a focused investigation of the impact of maps on navigation tasks of varying complexity. Prior art shows that use of map-like memory can aid performance in longer-horizon embodied navigation tasks, but focuses on approaches for constructing the maps from partial (visual) observations of the environment, and architectures that can make use of them. This Papers attempts to study _what_ information is most useful, assuming _perfect_ agent localizaion and map construction abilities: * what is a useful map representation? * benchmark agents with egocentric mapping, agents using oracle maps, and agents possessing no map memory. In order to do so the authors introduces the multiON task: an extension of the ObjectGoal navigation task, that involves navigation to an ordered sequence of objects placed within realistic 3D interiors. The task is about navigation to an episode-specific ordered list of objects: the episode-specificity requrie grounding object class labels to their visual appearance as well as memorizing previously encountered objects to be retrieved later. The paper is mostly an experimental paper built around benchmarking performance on variants of the multiOn task. The benchmarking is setup as experiments with variations of a base agent with access to a 'perfect' map of the environment. Notably, tthe environment has a reward structure that effectively guides the agents towards the tasks subgoals. Agents are all trained using the same prodece based on PPO, details in section 5.1. The papers verifies experimentally that: * spatial maps are indeed useful (agents with perfect map is consitently better performing) * goal location is much more useful than occupancy information * recognition of the goal from visual features and integration into a map memory is more effective than accumulating image features into a map The authors also maesure that: ** a simple semantic map agent outperforms more complex neural image feature map agents ** oracle map agents can also achieve relatively low performance and formulate a strong claim: * semantic maps are indeed highly useful for navigation, with a relatively naïve integration of semantic information into map memory outperforming learned map representations. Furthermore, the paper confirms that navigation is indeed hard an point to possible research directions, e.g. * improved learned modules for integrating egocentric information into map representations

Strengths: Significance and novelty of the contribution. This paper takes 1 question: 'what information is most useful, assuming perfect agent localizaion and map construction abilities?' and builds a clear experiment to shed light and provide clear signal on how it would make sense to think about mapping and navigation. This is not an easy task! Navigation is a very relevant topic in beeding edge AI, and it has subject of solid investigation for many years in very different disciplines - off the top of my head I can think of methods rooted in classisc computer vision (e.g. classic slam using structure from motion), deep reinforcement learning, and neuroscience (e.g. neural-slam). In order to make any significant progress one will have to make some assumptions and frame the work in the prior art (whcih is huge). I really liked the inroduction of the paper - as it set up the stagee for something like: 'here are all those experiments you have always wanted to do, but never actually had the time to do prorperly', since the premise was something like 'imagine we could learn mapping, what would it make sense to learn?'. Continues...

Weaknesses: Significance and novelty of the contribution. Some of the assumptions that the authors then chose to make made the paper a less interestsing than I had hoped. making the investigation very deep, but narrow: * ...that mapping means creating some form of quantized top-down view of the environment. I don't see why a human readable map should be in any way optimal for a learnt system. Other work indeed shows you can decode a such a map from a hidden representation, but the choice of representation / architecture is very important for agent performance. * using MapNet as the one example of fully learn system. I appreciate that this choice might be motivated by the fact that the agent uses a mapping represenation compatible with the others leveraged in the paper. At the same time I think it is fair to say thatt the conclusions the authors derive from the experiments are not warranted. * the unusual reward structure using r_closer - what happens without this signal, in a setting that is less artificial? Overall I think the overall impact of the paper is limited by the artifically rescricted scope of the experiments. In particular, the conclusions about the learnt system are not convincing, because of the choice of agent design and representation, and lack of comparisons with baselines in the literature - that is, I think the results we see are like this because of the choice of agent, rather that something fundamental you have found. I would love if the authors could convince me otherwise in the rebutttal.

Correctness: To the best of my understanding the exhecution of the expeirments in the paper is sound, and the methodology is described in such a way that they should be reproducible (furthermore, the authors provide code). As discussed in the previous section I question the overall significance of the experiments.

Clarity: I found the paper overall quite clear, in the following I list a few suggestions: Line 122: 'The goal objects are inserted into realistic 3D environments allowing for controlling the complexity of the navigation episodes and the goal object category.' It would be beneficial to explain what is meant for complexity. I assume this could be, for example. distance between targets, but it would be good to be explicit. Line 126: 'Task evaluation metrics are defined independent of reward formulations, allowing exploration of different reward structures and other methods such as imitation learning.' At this point in the paper the task evaluation metrics have not been described yet, and this sentence sets the expectation that multiple methods will be explored (they are not). Perhaps the authors could consider clarifying this straight away. Line 167-170.: I don't understand what the authors are trying to say.

Relation to Prior Work: The authors could consider providing a discussion of the specific choice of agent / architecture / mapping representation. This could include an verview of other ways to think about navigation and mapping; one example could be work inspired by neuroscience, for example something more neuroscience inspired like: Banino et al, Vector-based navigation using grid-like representations in artificial agents. where the agent learns a goal oriented policy without the distance reward signal used in this submission.

Reproducibility: Yes

Additional Feedback: I would like to thank the authors for their submission and look forward to discussion in the rebuttal. Typos: Line 21 assumtion->assumtions --- AFTER REBUTTAL Thanks for the thorough rebuttal. Updating my score to 6.


Review 3

Summary and Contributions: This paper introduces a new set of navigation tasks in 3D environments as well as an extensive study of how mapping capabilities can increase performance on those task. Specifically the set of task is 3D scanned environments in which goals are inserted and the task is for an agent to navigate to a sequence of goal and to issue a special Found action every time it is in the vicinity of the next target. The experiments consists of running a deep rl agent with mapping capabilities, with different ablations from full oracle map to learned map to no map at all. It provides a comprehensive story as to how much does this mapping capability is important and to what extend performance degrades w/o it or when it is imperfect / learned.

Strengths: The paper is clear, straightforward. The tasks are interesting and clean and the empirical evaluation is also straightforward. The empirical study and the ablations make sense, and the results are consistent with what one would expect. The code for the task and for the agent has been provided as supplementary material.

Weaknesses: Even though the experimental section seems well polished, I found it quite unsurprising and perhaps was lacking some novelty. The most sophisticated non-oracle agent is coming from another paper (MapNet), and the other agents studied in this paper either have access to oracles, or are a degradation from MapNet (which sometime surprisingly perform better). I was also personally not convinced about the fact that increasing m (the number of object to correctly find in a raw) is so interesting. Contrary to the author claim, I find it quite normal that performance would degrade dramatically with m, due to the termination of episode if incorrect Found is issued . Arguably, for any task, you can make the get-m-successes-in-a-row-or-die version, and it will be just exponentially harder than the original task. I don't think that tells you much about the task or about the method being used to solve it. That being said, the task is clean and interesting, and the different ablations makes sense so I still think it is a good paper overall, it is just that this emphasis on m is not very relevant in my opinion.

Correctness: Most of the claims of the paper seem quite plausible, the experimental section looks quite polished and all the code is provided as supplementary material.

Clarity: Yes the paper is very clear and well written.

Relation to Prior Work: Yes the prior work seems reasonably developed and the current contributions are clearly separated from it.

Reproducibility: Yes

Additional Feedback: l201: do you also mean that Found cannot be randomly issued? l207: do you mean r_subgoal is issued once when calling Found while being in vicinity? - I would be curious to see how much making episode terminate on incorrect Found is making the problem (artificially as argued above) harder? It would be interesting to see how OracleMap + OracleFound would perform on the m=3 case for example. - If I understand correctly the global position of the agent is not part of the map? I found it surprising? It might be one reason the OracleMap agent is not perfect because it probably needs a few frames first to infer its position before it can then take shortest path to goal? Might be interesting to have an additional run OracleMapOraclePosition to disentangle that effect as well?


Review 4

Summary and Contributions: This paper proposes a task framework for benchmarking object-seeking navigation agents on a controlled axis of complexity. Its primary contributions are the generalization of the typical point/object goal navigation tasks to a sequential navigation problem, the temporal length of which can implicitly control task complexity, and results of several major variants of navigation agents in order to demonstrate the importance of maps and the bounds on performance via oracles.

Strengths: This is an important work, demonstrating several useful and interesting findings. The first useful contribution is the task benchmark, which I expect will become a useful axis of evaluation for future goal-directed navigation work. The second is the interesting difficulty of the problem even under oracle mapping scenarios, and the surprising difference in performance between the learned vs hard-coded semantic feature-based maps.

Weaknesses: I have several concerns about this work. - Am I correct that the time limit for the task is constant, independent of the number of sequential objects to reach? This seems severely limiting, since surely the optimal path length will depend on how many stages of object reaching are required. Could the authors comment on this choice and whether they tried extending the time limit based on the number of objects, or perhaps based on the ground truth optimal path length? - The oracle maps appear to be presented in allocentric coordinates, i.e. independent of the agent's location. Did the authors evaluate against an egocentric oracle map where the agent's location is always at the center? It seems like this could improve performance. - The taxi/courier task in papers like [1] seems like an important precedent for this form of sequential goal-reaching problem - Nitpick: the time penalty of -0.01 in the reward function is probably unnecessary here: any time-discounted agent will already prefer shorter paths. [1] Mirowski, Piotr, et al. "Learning to navigate in cities without a map." Advances in Neural Information Processing Systems. 2018.

Correctness: The method is well-justified, but as mentioned above I have concerns about the validity of the claims and their supporting experimental methodology regarding the difficulty of the task as the number of sequential goals increase; might this just be due to the reduced time limit?

Clarity: The paper is well-written and the importance of studying this topic is made clear.

Relation to Prior Work: As mentioned above, variations on this sequential goal-reaching task appear in previous works and those should be mentioned. Otherwise the work is well-situated in the literature.

Reproducibility: Yes

Additional Feedback: Given my concerns above, I'm unable to recommend acceptance. However, if the authors have clarifying information from e.g. other experiments regarding the fixed time limit, then I would revise my decision to acceptance. ---- UPDATE ---- After considering the authors' response, I am revising my score up to a 6. My main concern was whether it is reasonable for the fixed time limit to be constant irrespective of the complexity of the task. The authors explained that the time limit, while fixed, is far higher than what is required for the task. As such, I have no fatal issues with the paper. I would request that the authors please clarify this in the text of the paper, and I would also like to hear a proposal for how the time limit could be specified once the tasks do start to get too hard for 2500 steps. For example, if I want to extend this benchmark up to 20 sequential objects, how should I set the time limit? Would 10x the optimal length be reasonable, or should it follow some other growth function?

[Author Response · NeurIPS 2020]

We thank the reviewers for their detailed and thoughtful comments. We are happy to see that reviewers found our
paper has 'strong significance' and 'novel insight into map architectures' (R1), 'provides a clear signal on how to think
about mapping and navigation' (R2), gives 'tasks that are interesting and clean' and an 'extensive study of mapping
capabilities' (R3), and is 'an important work, demonstrating several useful and interesting findings' (R4). We answer
questions by reviewers below and will add related feedback with suggested references and clarifications in the paper.

**R1, R2, R3: Other agent memory architectures?** Our focus was to analyze the impact of information in map
representations. We agree that discussing and benchmarking more agent architectures will strengthen our findings.
Following reviewer suggestions, we conducted experiments with SMT [17], EgoMap [6] and FRMQN [Oh et al., 2016].
Results are consistent with our earlier findings, with all three performing better than the simple RNN memory in `NoMap`
but worse than `ObjRecogMap`, with differences being more pronounced on 3-ON. As suggested by R2, we will add a
discussion of the choice of agent architecture and memory representation, including Banino et al.

| | SUCCESS (%) | | | PROGRESS (%) | | | SPL (%) | | | PPL (%) | | |
|---|---|---|---|---|---|---|---|---|---|---|---|---|
| | 1-ON | 2-ON | 3-ON | 1-ON | 2-ON | 3-ON | 1-ON | 2-ON | 3-ON | 1-ON | 2-ON | 3-ON |
| NoMap (RNN) | 62 | 24 | 10 | 62 | 39 | 24 | 35 | 13 | 4 | 35 | 21 | 14 |
| ProjNeuralMap | 65 | 30 | 12 | 65 | 44 | 27 | 37 | 20 | 5 | 37 | 28 | 14 |
| ObjRecogMap | **79** | **51** | **22** | **79** | **62** | **40** | **56** | **38** | **17** | **56** | **45** | **30** |
| SMT [17] | 63 | 28 | 9 | 63 | 44 | 22 | 48 | 26 | 7 | 48 | 36 | 18 |
| EgoMap [6] | 66 | 36 | 16 | 66 | 48 | 32 | 49 | 27 | 12 | 49 | 35 | 24 |
| FRMQN [Oh et al., 2016] | 62 | 29 | 13 | 62 | 42 | 29 | 50 | 24 | 11 | 50 | 33 | 24 |

**R1: Measure variance with multiple seeds?** We did not include this analysis in the paper since early experiments
showed low training variance, and reporting results with one training seed is common in embodied AI for naviga-
tion [11,17,21-23]. The standard deviation across 5 training runs on 3-ON of the `OracleMap` and `ProjNeuralMap`
agents is $0.38\%$ and $0.72\%$ for SPL, and $0.52\%$ and $0.99\%$ for PPL (all within $1\%$).

**R2: Human-readable top-down maps are not necessarily optimal for learning-based agents.** Top-down maps
are indeed one of many possible spatial representations. They are a good starting point as they are common in real
life, and humans are taught to understand and use them reliably. We do not mean to imply they are optimal for
learning-based systems. Top-down maps do bring the advantage of a good abstraction, human interpretability, and they
impose inductive biases tied to the spatial structure of interiors that have been shown to outperform implicit memory
architectures in a variety of navigation tasks [6,11,12]. We will clarify this in the paper.

**R3,R4: Hardness of the task as $m$ increases and allotted time limit.** Reviewers asked whether it is surprising that
agents fail to perform well with increasing $m$ as: 1) the task is obviously exponentially harder, and 2) there is a
fixed max time limit. We clarify here and will add discussion of these points to the paper. **R3: Obvious that task is**
**exponentially harder?** With exponential decay, we expect the `OracleMap` SUCCESS rate to be $0.94$, $0.94^2 = 0.88$ and
$0.94^3 = 0.83$ as $m$ is increased from 1 to 3. The actual SUCCESS rate is much lower indicating room for improvement.
In addition, if the agent had the ability to remember past observations, finding the third goal should be easier if it was
encountered while looking for the first two goals. This is supported by the analysis in Section 1.4 of the supplement.
**R4: Is max time limit independent of $m$ in $m$-ON?** Yes, we set the max number of steps for all tasks to 2500 steps.
Nearly all ($> 99.5\%$) episodes terminate by calling the FOUND action rather than by reaching this limit. The mean and
median episode lengths being 276.2 and 151 steps respectively for 3-ON experiments. We verified that this fixed max
step threshold has negligible impact on the results reported in the paper.

**R3: How much is episode terminating on incorrect FOUND making the problem harder?** Yes, the FOUND action
makes the problem much harder. Without it, the task is reduced to a search problem bounded by the max step limit. We
evaluated an `OracleMap` agent with 'OracleFound' (FOUND called automatically). This achieves SUCCESS of $90\%$ in
3-ON, compared to $48\%$ for `OracleMap` that must call FOUND. The FOUND action follows the recommendation of [2]
and is a realistic requirement: it indicates the agent correctly identified the object and is ready to move on to the next
goal. In initial experiments, we trained agents with the episode not ending on incorrect FOUND. These agents fail to
learn, issuing FOUND randomly and wandering in an episode until the max step limit is reached.

**R2: Unusual 'r_closer' (reward for coming closer to goal).** In early experiments without this reward, agents failed
to train even after 40 million steps. This form of reward shaping is commonly employed in RL-based embodied
navigation [31,37], and is crucial for facilitating training as a weak supervisory signal that mitigates sparse rewards.

**Other clarifications.** R3: Global position of agent is not part of the map? and R4: Oracle maps appear to be in
allocentric coordinates? As input to the agent, all map-equipped agents receive maps centered and rotated at the agent's
location and orientation. R4: Taxi/courier task an important precedent. Thank you for the suggestion, we will add this
work to the discussion. R2,R3: We will clarify the exposition for L122, L126, L167-170, L201, and L207.

[Meta-Review · NeurIPS 2020]

The paper had initially received 4 mixed reviews, with two reviewers recommending acceptance and two reviewers recommending rejection. Three of the four reviews where borderline. The reviewers agreed on most points, in particular the strong experimental nature of the paper, which does not propose a new method for the targeted navigation problem, but tests the impact of ego-centric birds-eye representations and projective mapping. The reviewers also agreed that the performed experiments were interesting and important for the field of embodied vision / robotics. The question was therefore, whether the results of the experiments were strong enough and whether they provided enough insights, i.e. whether the were representative for a sufficiently large class of problems. Minor weaknesses perceived by some of the reviewers were the artifically rescricted scope of the experiments. The author's response could address several remarks and clarify misunderstandings, and the discussion between the reviewers (and the AC) lead to a more favorable assessment of the paper. The reviewers judged that the experimental results are important for the field, the AC concurs.